# Manipulating the Epigenome in Nuclear Transfer Cloning: Where, When and How

**DOI:** 10.3390/ijms22010236

**Published:** 2020-12-28

**Authors:** Kilian Simmet, Eckhard Wolf, Valeri Zakhartchenko

**Affiliations:** 1Institute of Molecular Animal Breeding and Biotechnology, Gene Center and Department of Veterinary Sciences, LMU Munich, 81377 Munich, Germany; k.simmet@gen.vetmed.uni-muenchen.de (K.S.); ewolf@genzentrum.lmu.de (E.W.); 2Center for Innovative Medical Models (CiMM), LMU Munich, 85764 Oberschleißheim, Germany; 3Laboratory for Functional Genome Analysis (LAFUGA), Gene Center, LMU Munich, 81377 Munich, Germany

**Keywords:** nuclear transfer, reprogramming, epigenetics, genome editing, epigenome editing

## Abstract

The nucleus of a differentiated cell can be reprogrammed to a totipotent state by exposure to the cytoplasm of an enucleated oocyte, and the reconstructed nuclear transfer embryo can give rise to an entire organism. Somatic cell nuclear transfer (SCNT) has important implications in animal biotechnology and provides a unique model for studying epigenetic barriers to successful nuclear reprogramming and for testing novel concepts to overcome them. While initial strategies aimed at modulating the global DNA methylation level and states of various histone protein modifications, recent studies use evidence-based approaches to influence specific epigenetic mechanisms in a targeted manner. In this review, we describe—based on the growing number of reports published during recent decades—in detail where, when, and how manipulations of the epigenome of donor cells and reconstructed SCNT embryos can be performed to optimize the process of molecular reprogramming and the outcome of nuclear transfer cloning.

## 1. Introduction

The genome sequence includes the principal instructions to build, develop, and maintain an organism, but epigenetic mechanisms determine how this information is used in specific cell types during development and differentiation as well as in physiological and pathological processes. Whereas the DNA sequence is almost identical in all cells of an organism, epigenetic determinants—summarized as the “epigenome”—vary widely across different cell types and developmental stages and thereby modulate the cellular gene activity profile. Epigenetic mechanisms, i.e., DNA methylation, histone protein modifications, and effects associated with non-coding RNAs, are influenced by endogenous physiological and pathological stimuli, but also by exogenous environmental effects. Experimental studies in model organisms, but also epidemiological studies in humans indicate that epigenetic mechanism are—at least in part—heritable (reviewed in [1]).

Nuclear transfer (NT) experiments were originally designed to answer one of the most intriguing questions of developmental biology: is embryonic development and differentiation and subsequent fetal growth associated with irreversible modifications in the resulting somatic cells? NT experiments were first performed in amphibians, where transfer of a differentiated cell into an enucleated oocyte resulted in the development of an adult animal, demonstrating that totipotency of somatic cells can be restored [2]. In mammalian species, the breakthrough result of somatic cell nuclear transfer (SCNT) was the sheep Dolly, which was produced by transfer of an adult mammary epithelial cell to an enucleated oocyte [3]. Over the last decades, SCNT became a standard methodology and there are now thousands of clones around the world including 22 mammalian species: sheep, cow, goat, pig, mouse, rat, rabbit, dog, wolf, domestic and wild cat, mouflon, mule, buffalo, horse, gaur, red deer, ferret, camel, ibex goat, coyote, and macaque monkey (reviewed in [4,5]).

Cloning offers great perspectives for agricultural and biomedical applications as well as for basic research. In agriculture, cloning provides great possibilities to rescue endangered species, to protect the genetic resources of commercially important species, and to accelerate the propagation of breeding livestock (reviewed in [6,7]). In combination with genome editing, SCNT can produce animals with desirable traits including rapid growth, disease resistance, and improved product quality [8,9,10]. In biomedicine, the generation of genetically engineered animals with various features relies on SCNT, including donors for organ xenotransplantation [11,12], models for human diseases [13,14,15], and living bioreactors to produce compounds for diagnostics and therapy [16]. SCNT can also be used to generate isogenic embryonic stem cells (ntESCs), especially human ntESCs, thus providing an important source for organ regeneration [17]. In basic research, SCNT is an excellent model for understanding how cell memory can be fully reprogrammed to generate totipotent cells [18] and to perform hypothesis-driven developmental studies in model organisms other than mouse [19].

However, practical application of SCNT technology is hindered by its low efficiency. Losses occur throughout preimplantation, postimplantation, and perinatal development. Surviving animals often exhibit various abnormalities, such as large offspring syndrome (LOS), enlarged placentas and organ defects (observed in cattle, sheep and mice), or obesity in mouse and abnormal teat numbers and cleft lips in pigs [4,20,21,22,23]. The causes of these abnormalities can be divided into four main categories: trauma during micromanipulation, insufficient reprogramming competence of the used oocytes, resistance to reprogramming of the used donor nuclei, and anomalies induced by in vitro culture of the reconstructed SCNT embryos. These factors may result in abnormal epigenetic profiles and gene expression patterns in cloned embryos, which are considered to be the main barriers to normal development [5,18,24].

Epigenetics refers to the modulation of gene expression through the physical and biochemical properties of chromatin without changing the DNA sequence. Epigenetic mechanisms include DNA methylation, post-translational modification of DNA-binding proteins and the integration of chromatin-binding proteins to maintain chromatin in either an active or repressed configuration [25]. During normal mammalian preimplantation development, fundamental epigenetic changes take place to ultimately generate an organism from two differentiated gametes. The entire DNA methylome is erased—except for imprinted regions—and later re-established [26]. For successful SCNT, the pattern of epigenetic modifications in the differentiated nucleus of the donor cell must undergo remodeling to become like the pattern present in the nucleus of a zygote. Incomplete epigenetic remodeling and aberrant patterns of DNA methylation or histone acetylation in SCNT embryos have been identified in numerous studies and all contribute to the inefficiency of SCNT [24,27,28,29].

Many attempts have been made to improve the development of SCNT embryos by manipulating their epigenome. In this review, we describe and discuss in detail where, when and how these manipulations may occur and show examples of global and specific manipulations of the epigenome of nuclear donor cells or SCNT embryos from the growing number of reports published during the last decades. Finally, we discuss the future perspectives of manipulating the epigenome for improving cloning efficiency.

## 2. Epigenetic Modulation of Gene Expression

DNA methylation within gene promoter regions is associated with a repressive transcriptional state and plays a pivotal role in mammalian development. The most prominent methylation site is the 5th carbon of cytosine (5mC). DNA methylation is established and maintained by DNA methyltransferases (DNMTs). Demethylation occurs by ten-eleven translocation (TET) protein-mediated oxidation [30].

Histone proteins are key players in nucleosome formation and DNA packaging. They are subject to many post-translational modifications affecting mainly the N-terminal tails of core histones, including acetylation (lysine), methylation (lysine and arginine), phosphorylation (serine and threonine), ADP ribosylation, sumoylation (lysine), ubiquitylation (lysine), butyrylation, citrullination, crotonylation, formylation, proline isomerization, and propionylation [31].

Acetylation of histone tails loosens the histone-DNA interactions and enables gene expression, whereas deacetylation of histone tails strengthens the interactions and is generally associated with gene repression. Histone acetylation is regulated by the balanced action of histone acetyltransferases (HATs) and histone deacetylases (HDACs) [32].

The effect of histone methylation depends on both the modified residue and the extent of methylation. Histone lysine (K) methylation can exist in one of three states: mono-, di- or tri-methylation. Di- and tri-methylations at H3K4, H3K36, and H3K79 are typically gene-activating, with H3K4 tri-methylation (H3K4me3) marking promoters, and H3K36 and H3K79 methylations occurring primarily over gene bodies. Mono-methylation of H3K4 is an activating mark unique to enhancers. H3K9 and H3K27 methylations are generally gene-repressive, but serve unique functions. H3K27me3 is considered easily reversible and marks dynamically regulated genes, rendering it especially important in development, when genes need to be switched on and off in a highly dynamic fashion depending on developmental signals. H3K9me3 is characteristic of heterochromatin, whereas H3K9me2 is found more commonly at silent or lowly expressed genes in euchromatin. Histone K methylation is regulated by histone methyltransferases and histone demethylases (reviewed in [33]).

## 3. Epigenetic Abnormalities in Embryos, Fetuses and Offspring Derived by SCNT

Embryos, fetuses and offspring generated by SCNT may suffer from a variety of epigenetic abnormalities, which are attributed to insufficient or aberrant epigenetic reprogramming of the somatic donor nucleus. These epigenetic barriers to SCNT cloning are summarized in Table 1.

The process of nuclear reprogramming by the recipient cytoplasm can be characterized as a conflict between the cytoplasm of oocytes and the donor cell nucleus, when the transplanted nucleus is resistant to the reactivation of genes necessary for early development, and switching off genes expressed in its former state is hampered (reviewed in [34]). This conflict may cause abnormal gene expression in cloned embryos due to alterations in embryonic genome activation (EGA) and in the degradation of maternal transcripts from the oocyte [35]. The tightly regulated process of maternal-to-embryonic transition during early development of the fertilized oocyte, which is timed in a species-specific manner (reviewed in [36,37]), may be difficult to mimic by the SCNT technology. Specifically, disturbed transcription by RNA polymerase I [38] and failure in reprogramming specific DNase I hypersensitive sites of somatic donor nuclei, which prevents the binding of chromatin remodeling factors to regulate gene expression in cloned embryos [39], have been observed. Studies of cloned mouse embryos revealed continued expression of some somatic genes [40] and failed activation of important pluripotency genes such as *Oct4* [41] and *Sox2* [42] even at the blastocyst stage.

In terms of epigenetic marks, SCNT embryos frequently fail to fully undergo the wave of demethylation observed during normal embryonic development, resulting in increased DNA methylation levels compared to embryos derived by fertilization [27,43,44,45]. Rapid deacetylation of histones as well as abnormal patterns of histone methylation in cloned embryos are other consequences of SCNT [46,47]. Several histone variants also exhibit abnormalities, such as the delayed change of H1FOO (oocyte-specific H1) to somatic H1s [48] and the replacement of repressive H3 in donor cell nuclei by maternal H3.3 [49]. In addition to abnormalities in epigenetic marks, alterations in higher-order chromatin structure were noted in SCNT embryos [50].

Analyses of transcriptome and epigenetic changes during SCNT reprogramming using recently developed low-input RNA sequencing techniques have revealed molecular defects and provided approaches to overcome critical barriers to epigenetic reprogramming [51,52,53,54].

A high proportion of SCNT embryos fail to implant. Studies in bovine showed that the endometrium responds differently to cloned embryos as compared to embryos produced by in vitro fertilization (IVF), indicating abnormalities in embryo-maternal communication and pregnancy recognition signaling [77,78]. Subsequent studies revealed dysregulation of genes involved in cell signaling and placental development [79,80].

Many of the pathologies observed in cloned conceptuses reflect problems with placental function. Among the earliest abnormalities in cloned blastocysts is DNA hypermethylation in the trophectoderm. Several imprinted genes have been found to be normally expressed in cloned fetuses but abnormally expressed in the placentas. X chromosome inactivation (XCI) seemed to be normal in the embryo proper, while aberrant expression of X-linked genes has been observed in the placenta (reviewed in [81]). The authors concluded relatively normal reprogramming in the embryonic lineage of cloned embryos but aberrant reprogramming in their trophectoderm. In relation to this, abnormal expression of *Xist* has been documented in cloned embryos of both sexes, leading to a decreased expression of X-linked genes and abnormal development [82].

Abnormalities in DNA methylation of SCNT embryos during preimplantation development may be maintained throughout development and were discovered in cloned bovine fetuses [83]. Significant DNA hypermethylation was detected in liver tissue of cloned bovine fetuses and correlated with fetal overgrowth [84]. Alterations in DNA methylation levels, including hypermethylation and hypomethylation, either global or at specific gene sequences, have been observed in abnormal or dead bovine SCNT fetuses or calves, compared with either conventionally produced controls or apparently normal clones [83]. Even phenotypically healthy bovine clones showed DNA hypermethylation and a much higher variability in DNA methylation levels compared to monozygotic twins generated by embryo splitting [85].

Abnormal patterns of gene expression observed in preimplantation SCNT embryos may persist throughout fetal development up to birth. Genes aberrantly expressed in blastocysts were also aberrantly expressed in the organs of clones that died shortly after birth (reviewed in [86]).

## 4. Non-Specific Modulators of the Epigenome

The inhibition of DNA methyltransferases using chemical compounds (DNMTi) targets the entire chromatin landscape and globally reduces the amount of repressive DNA or histone methylation marks. Although effectively inducing hypomethylation in donor cells, the DNMTi 5-aza-2-deoxycytidine (5-aza-dC) had no beneficial effect on the development of cloned embryos when applied during embryo culture [87,88,89,90]. In contrast, treatment of donor cells with another globally acting hypomethylating agent—S-adenosyl homocysteine (SAH)–significantly improved the development of bovine SCNT embryos. This was attributed to a lower cytotoxicity of SAH allowing the use of higher concentrations and to SAH mediated demethylation of one X chromosome and increased telomerase activity [91].

Other globally acting agents are histone deacetylase inhibitors (HDACi). They prevent the removal of acetyl groups from lysine residues of histone proteins and thus maintain a gene expression permissive histone mark. Trichostatin A (TSA) is a prominent HDACi, which has been mostly used to treat cloned embryos. It improved cloning efficiencies in several species, including mouse [29,92,93], cattle [22,94,95], pig [96,97], and rabbit [98]. Treatment of cloned embryos with TSA facilitated serial recloning of mice for up to 25 generations [99]. Nevertheless, other studies reported no improvement of TSA regarding full-term development of bovine SCNT embryos [100,101] or even detrimental effects on rabbit SCNT embryos [102], which could be due to ineffective or toxic doses of TSA, respectively. TSA is known to be teratogenic [103] and can result in a significant reduction of embryo development [104,105] as well as severe placentomegaly [93] when the concentration is too high or the exposure too long. The effect of TSA treatment of somatic donor cells has only been investigated in bovine and increased cloning efficiencies at adequate concentration and duration of treatment. Treated cells were synchronized at G0/G1 stage and showed hyperacetylation of H3K9 as well as decreased DNA methylation levels [88,90,94,106,107].

Further studies have refined earlier efforts and applied the less-toxic HDACi Scriptaid (6-(1,3-dioxo-1H, 3H-benzo[de]isoquinolin-2-yl)-hexanoic acid hydroxyamide). At appropriate doses and exposures times, it improved the cloning efficiency of highly inbred miniature pigs [47,108] and of inbred mice [104].

The HDACi compounds suberoylanilide hydroxamic acid (SAHA) and oxamflatin improved the full-term development of cloned mice [109]. This group treated cumulus cell-derived mouse SCNT embryos with SAHA or TSA and achieved up to 16% development to term. In pig cloning, treatment of SCNT embryos with SAHA or 4-iodo-SAHA after fusion and activation resulted in healthy pigs from donor cells that had a particularly high rate of postnatal mortality when using Scriptaid [110]. Treatment of cloned mouse embryos with psammaplin A (PsA), another HDACi, increased full-term development four-fold when cytochalasin B (CB) was used during activation. CB prevents pseudo-second polar body extrusion by inhibiting actin polymerization, which can be also achieved using latrunculin A (LatA). Interestingly, the combination of PsA and LatA was more potent, increasing development 11.5-fold [111].

A non-chemical approach to modulate the donor cell’s epigenome relies on the manipulation of their metabolism. Folic acid (folate) is a critical player in methylation reactions. Deprivation of folate altered the DNA methylation pattern of bovine fetal fibroblasts, resulting in two-fold improved development to blastocyst when these cells were used for SCNT. Gene expression and epigenome signatures of SCNT blastocysts from folate-deprived donor cells were more similar to blastocysts derived by in vitro fertilization than those of control SCNT blastocysts from folate-exposed donor cells [112].

## 5. Specific Attempts to Modulate the Epigenome

### 5.1. Manipulation of Methyl-CpG-Binding Domain Proteins and Transcription Factors

Methyl-CpG-binding domain proteins (MBPs) connect DNA methylation to histone modification and change fundamentally during somatic cell reprogramming. Overexpression of methyl-CpG–binding protein 2 (MECP2) in mouse donor cells increased the oxidation of 5-methylcytosine (5mC) to 5-hydroxymethylcytosine (5hmC), the expression of pluripotency genes, and the developmental capacity of cloned blastocysts [113]. The authors speculated that MECP2 activates TET3, which contributes to demethylation of both the paternal and maternal genome [114]. Similar to MECP2, overexpression of TET3 in donor cells also resulted in an increased level of 5hmC and expression of pluripotency genes along with an improvement of the development of goat [115] and bovine [116] SCNT embryos. These studies indicate that TET3 activity is crucial for reprogramming after SCNT and for the development of the resulting cloned embryos.

Overexpression of methyl-CpG-binding domain protein 3 (MBD3), a core component of the nucleosome remodeling and deacetylase complex, in porcine SCNT embryos increased blastocyst rate and decreased the DNA methylation of *NANOG*, *OCT4*, and *LINE1*, and thus upregulated their expression levels close to those found in in vivo fertilized embryos [117].

Transient overexpression of the double homeobox transcription factor (DUX), a key inducer of EGA, in cloned mouse embryos improved their full-term development. Moreover, transcriptome profiling revealed that DUX expressing cloned embryos are similar to fertilized embryos. Furthermore, overexpression of DUX combined with knockdown of DNMTs promoted the full-term of cloned embryos [118].

Hypoxia inducible factor 1 subunit alpha (HIF1A), a transcription factor that allows for cell survival at low oxygen tension, promotes a metabolic switch from somatic cell specific oxidative phosphorylation to glycolysis used by early embryos [119,120]. Stabilization of HIF1A by treatment of donor cells with cobalt chloride (CoCl_2_) upregulated mRNA abundances of glycolytic enzymes and improved development of porcine cloned embryos to the blastocyst stage [121]. Shifting the metabolism of donor cells toward glycolysis can thus be a simple way for improving cloning efficiency.

### 5.2. Transcriptional and Epigenetic Modulation of Xist

X chromosome inactivation (XCI) is a mechanism of dosage compensation, where one X chromosome is transcriptionally silenced in every diploid cell of a female organism during early embryonic development. Untranslated *Xist* RNA originating from the X chromosome that will be inactivated coats the chromosome and thus leads to silencing [122]. Increased expression of *Xist* from the active X chromosome has been documented in cloned embryos of both sexes, leading to a decreased expression of X-linked genes and abnormal development [82]. Therefore, blocking of abnormal *Xist* expression using small interfering RNAs (siRNAs), knockout of the maternal *Xist* allele, or epigenetic modification of the *Xist* locus are strategies to improve development of cloned embryos.

Blocking of *Xist* expression via injection of an *Xist*-specific siRNA into early male mouse SCNT embryos resulted in a 10-fold increased blastocyst rate and an increased rate of development to term [123]. However, the knockdown of *Xist* could only enhance the developmental competence of male but not female mouse cloned embryos; it was hypothesized that in the latter siRNA injection did not consistently reduce *Xist* expression to normal levels [124]. On the other hand, knockout of *Xist* on the active X chromosome normalized *Xist* expression in cloned embryos, leading to remarkable improvements in birth rates of both male and female offspring [61].

Recently, Yang, et al. [125] demonstrated that injection of an anti-*XIST* shRNA expression plasmid but not anti-*XIST* siRNA at the two-cell stage reduced *XIST* RNA levels at the blastocyst stage and enhanced developmental ability of male pig SCNT embryos. This was most likely due to a prolonged gene silencing effect of plasmid-expressed shRNA (over five days vs. 2–5 days with siRNA). Knockout of *XIST* in male porcine donor cells resulted in suppression of ectopic *XIST* expression and a global reduction of H3K9me3. The quality of preimplantation stage SCNT embryos and their development to term was significantly improved [62].

A recent study in mouse embryonic fibroblasts (MEFs) demonstrated an epigenetic role of *Xist* that is independent of *Xist* expression. Zhang, et al. [126] tested transcriptional-activator-like effector-based designer transcriptional repressors (R-dTFs) and activators (A-dTFs) for several regions of the *Xist* gene. An R-dTF specific for the *Xist* inton 1 enhancer region did not alter *Xist* expression, but improved the generation of induced pluripotent stem cells (iPSCs) and the development of SCNT embryos from MEFs expressing this R-dTF. These effects were more pronounced with male than with female donor MEFs. In contrast, expression of an A-dTF specific for the same region of *Xist* decreased the success of iPSC generation and development of SCNT embryos. The positive effect of the *Xist* intron 1 R-dTF was explained by a local enrichment of H3K9me3 followed by X-chomosome opening, repression of X-linked genes and eventually the activation of pluripotency genes [126].

### 5.3. Modulation of Histone Methylation

In cloned mouse embryos, reprogramming resistant regions (RRRs) high in H3K9me3 were identified, which could be reactivated by overexpression of *Kdm4d* encoding lysine demethylase 4D, simultaneously improving SCNT efficiency [52]. In a single-cell RNA-sequencing approach, Liu, et al. [51] identified inactivation of *Kdm4b* and *Kdm5b* (encoding demethylases preferentially acting on H3K9me3/2 and H3K4me3/2/1, respectively) as causal for developmental arrest of mouse SCNT embryos at the two- and four-cell stage, respectively. Co-injection of *Kdm5b* and *Kdm4b* mRNAs into the recipient oocytes before SCNT restored the transcriptional profiles of cloned embryos and greatly improved blastocyst rate to over 95%, as well as the production of cloned mice.

Modulation of H3K9 methylation was also used to improve the efficiency of SCNT in bovine. Liu, et al. [54] reported that global hypermethylation of H3K9 in bovine eight-cell SCNT embryos is linked to a deficient expression of two H3K9-specific demethylases, KDM4D and KDM4E. Overexpression of the more crucial KDM4E normalized the transcriptome profile of SCNT embryos and improved cloning efficiency, indicating that KDM4E is an essential epigenetic regulator of embryonic genome activation and that its deficiency in SCNT embryos results in persistent H3K9me3/2 barriers to successful reprogramming.

Another study found H3K4me3 hypermethylation and an increase in 5mC/5hmC as well as an abnormal transcriptional profile in bovine SCNT embryos. Injection of H3K4me3-specific demethylase 5B (KDM5B) encoding mRNA increased the blastocyst rate significantly and rescued transcription of aberrantly silenced genes while the memory of past donor cell transcriptional activity was repressed [56].

H3K27me3 in the donor cell chromatin is an epigenetic barrier to EGA in SCNT embryos. Yang, et al. [127] observed that *Kdm6a* and *Kdm6b* (encoding H3K27me3-specific demethylases) were not adequately activated in cloned mouse embryos. Using donor cell lines with fluorescent reporter genes, the authors addressed the question if supplementation of these KDMs can expedite EGA and improve development of mouse SCNT embryos. The injection of *Kdm6a* mRNA into enucleated oocytes improved EGA and preimplantation but not full-term development of SCNT embryos. In contrast, injection of *Kdm6b* mRNA had a negative effect on development. Interestingly, knockdown of *Kdm6b* (which resulted in increased *Kdm6a* mRNA levels) not only facilitated EGA and improved development to the blastocyst stage, but also increased development to offspring [127].

Injection of an antibody against the H3K9/H3K27 methyltransferase EZH2 into recipient oocytes reduced the H3K27me3 levels of porcine SCNT embryos and improved their development significantly [128].

A different approach to modulate the histone methylation pattern is the use of chemical inhibitors of histone methyltransferases, such as GSK126 for EZH2, BIX-01294 for the H3K9 methyltransferase G9A, or chaetocin for the suv39 family of H3K9 methyltransferases. Incubation of porcine SCNT embryos with these compounds reduced the levels of the respective epigenetic marks and improved cloning efficiency [128,129,130,131]. Incubation of cloned mouse embryos with GSK126 or BIX-01294 corrected some abnormal epigenetic modifications, but had no effect on preimplantation development [132].

### 5.4. Modulation of Genomic Imprinting

Genomic imprinting, which is the silencing of one parental allele mediated by DNA methylation and histone modifications, plays a crucial role in fetal growth and development [81]. In SCNT, a great proportion of cloned embryos is lost after implantation. Complete loss of H3K27me3 imprinting was found in mouse preimplantation cloned embryos [53]. However, loss-of-imprinting of H3K27me3 genes was not observed in porcine and bovine post-implantation cloned embryos, indicating that the H3K27me3-imprinting system may not be conserved across species [74]. Loss-of-imprinting in *Sfmbt2* was found to contribute to the placenta overgrowth phenotype of cloned mouse embryos, while *SFMBT2* is not imprinted in pig, bovine, or human [133].

Monoallelic deletion of four H3K27me3-imprinted genes (*Sfmbt2*, *Jade1*, *Gab1*, and *Smoc1*) in donor cells normalized their expression patterns in mouse SCNT embryos, increased the cloning efficiency to 14%, and prevented placental defects and fetal overgrowth. Among the four genes, deletion of *Sfmbt2* was the most effective in improving SCNT efficiency [134]. Deletion of the entire *Sfmbt2* miRNA cluster improved the birth rates of clones more than twofold and ameliorated placental overgrowth [75].

Silencing of the retrotransposon-derived imprinted gene *RTL1* was suggested as a principal cause of pregnancy failure after transfer of SCNT embryos to recipients [79,135]. Restoration of *RTL1* expression in pig donor iPSCs rescued the loss of cloned fetuses [74].

Primordial germ cell 7 (*PGC7*), a gene that maintains the methylation levels of imprinted genes, is often abnormally imprinted in cloned embryos. Overexpression of *PGC7* in fetal goat donor cells corrected the expression levels of the insulin-like growth factor 2 receptor (*IGF2R*) gene and of *XIST* in SCNT embryos, which significantly improved development to live offspring [76].

### 5.5. Transcriptional Manipulation and Epigenome Editing Using dCas9

The genome editing toolbox of Cas9 nuclease and clustered regularly interspaced palindromic repeats (CRISPR) did not only enable researchers to edit the genome precisely. Methods have been developed to also repress or activate transcription as well as to edit the epigenome. Catalytically inactive Cas9 (dCas9), that cannot induce DNA double strand breaks, is able to repress transcription at a specific locus by binding to the DNA target sequence. When dCas9 is fused to a transcriptional regulator, repression is either enhanced or, in contrast, transcription is activated. Precisely editing the epigenome is enabled by dCas9 fused to methyltransferase or the catalytic domain of TET proteins, depositing or removing DNA methylation marks, respectively. Additionally, histone modifications may be altered by a versatile array of tools, depositing or removing histone methylation and acetylation (reviewed in [136]). Epigenome editing in mouse oocytes was successfully used to manipulate coat color-related phenotypes and to correct imprinting disorders [137]. In addition, the first imprinting disease models were created by targeted DNA demethylation in zygotes [138]. Many target genes, whose repression or activation may improve the efficiency of SCNT, are known and inducible CRISPR/dCas9 targeting approaches allow their modulation in a spatially and temporally controlled manner. Precise epigenome editing using dTFs has been performed to modify the *Xist* locus in fibroblasts (see Section 5.2) [126] and is an excellent example, of how the more flexible and easier-to-handle dCas9 technology can influence reprogramming procedures.

## 6. Combined Approaches for Manipulating the Epigenome

It is reasonable to speculate that simultaneous manipulation of several key players in the process of nuclear remodeling and reprogramming using a combination of approaches might provide a better option for removing multiple epigenetic barriers and for improving the developmental competence of cloned embryos. The combined manipulation or treatment can be performed by various means, including treatment of donor cells and/or cloned embryos with different compounds affecting more than one epigenetic modification simultaneously to obtain a synergistic effect on the development of cloned embryos (Figure 1).

In rabbits, treatment of cloned embryos with the combination of two HDACis, TSA and Scriptaid, was more beneficial than the use of a single HDACi for improving cloning efficiency [139].

Treatment of donor cells with 5-aza-dC and TSA improved cloning efficiency in pigs [140], but not in cattle [100]. Such treatment for both donor cells and early cloned embryos was beneficial in bovine [141] and buffalo [142] cloning.

Combination of DNMTi and HDACi, RG108 and Scriptaid [143], Zebularine and Scriptaid [144], as well as BIX-01294 and Scriptaid [129] or TSA [145] improved porcine and sheep cloning efficiency after treatment of donor cells or cloned embryos. The expression levels of *OCT4*, *SOX2*, *H19*, *IGF2*, and *DNMT1* genes in treated cloned embryos were more similar to IVF embryos than without treatment.

*Xist* KO donor cells coupled with *Kdm4d* mRNA injection resulted in very high efficiencies (up to more than 20%) of mouse cloning with different types of donor cells. However, many of the cloned embryos still suffered from postimplantation developmental arrest and surviving embryos had an abnormally large placenta [55].

A study of Mizutani, et al. [146] presents an exciting example for cloning mice from “unclonable” adult neurons with the combination of two epigenetic approaches, the use of donor cells with reduced amounts of repressive epigenetic marks and treatment of cloned embryos with HDACi. First, using a specific antibody, they identified cells with reduced amounts of the repressive histone mark H3K9me, i.e., CA1 pyramidal cells in the hippocampus and Purkinje cells in the cerebellum. After SCNT, the reconstructed embryos were treated with TSA. Using CA1 cells, cloned offspring were obtained at high rates, reaching 10.2% and 4.6% for male and female donors, respectively. This study demonstrated that reduced amounts of H3K9me2 and increased histone acetylation act synergistically to improve the efficiency of SCNT.

One of the most efficient mouse cloning protocols used the combination of TSA and antioxidant vitamin C in culture medium with deionized bovine serum albumin. This resulted in activation of reprogramming-resistant genes, demethylation of H3K9me3/2/1 and in 15% of cloned embryos developing to term, indicating that this treatment can overcome major epigenetic barriers to successful nuclear reprogramming [147].

Aberrant DNA methylation as a critical epigenetic barrier to mouse cloned embryo development can be rescued by inactivation of DNMTs, resulting in improved cloning efficiency, which is further enhanced by simultaneous removal of additional epigenetic barriers. Combining inhibition of DNMTs by siRNAs for *Dnmt3a* and *Dnmt3b* with overexpression of histone demethylases by injecting enucleated oocytes with *Kdm4b* and *Kdm5b* mRNAs led to stronger reductions in inappropriate DNA methylation and synergistic enhancement of full-term development of cloned embryos [63].

Cynomolgus monkeys (*Macaca fascicularis*) have been successfully cloned by NT using fetal fibroblasts and injection of H3K9me3 demethylase and *KDM4D* mRNA together with TSA treatment at the one-cell stage [148].

## 7. Concluding Remarks and Future Perspectives

Manipulating the epigenome of donor cells or cloned embryos by a single approach or by the combination of several approaches can overcome some biological barriers to epigenetic reprogramming during SCNT resulting in a significant improvement of this process. Czernik, et al. [149] argued that the most promising strategies are those acting on the entire genome, such as the forced expression of histone demethylases or conversion of the chromatin structure typical for somatic cells to a spermatid-like structure. However, none of the approaches seems to be perfect due to the inherent dynamic nature of epigenetic modifications. Epigenetic states, once corrected, may revert to the original state because of the reversible nature of epigenetic modifications [150]. Some genes that escape or resist reprogramming may not respond to epigenome modifying agents. Specific epigenetic modifications, such as repressive histone lysine methylation marks, are very stable and difficult to reprogram; alterations may not persist in the reconstructed embryos and can be rapidly restored to the levels in donor cells [151].

Pharmacological tools currently available for manipulation of the epigenome operate globally at their target enzymes and can generate significant side effects. In addition, these drugs (i.e., HDACi and DNMTi) may silence as many genes as they activate, likely due to direct and indirect effects on other transcriptional regulators and cell signaling pathways [152,153].

A second group of tools involves the use of traditional genetic knockout/knock-in, transgenic, viral, and/or RNA interference (RNAi) technologies to manipulate specific epigenetic enzymes. While these approaches typically allow improved substrate specificity, including isoforms or subclasses and even limited cellular specificity, they still operate universally within a given cell and therefore lack the ability to modulate the epigenome in specific ways [154]. Thus, the major shortcoming of current pharmacological and genetic tools is that they lack the specificity to direct epigenetic changes at specific sequences within DNA, or even at specific genes.

New tools and strategies to promote gene-specific epigenetic modifications, referred to as epigenome editing, are now available and open the perspective of precisely targeting epigenetic deregulations. This allows the development of new hypotheses regarding epigenetic function and drastically reduces undesired side effects. Epigenome editing would perfectly complement recent developments in genomic, proteomic, and metabolomic profiling of embryos to link their viability and reproductive potential to specific signatures. Application of spectroscopy and bioinformatics for noninvasive metabolomic profiling of embryo culture media revealed a unique footprint for embryos with high reproductive potential compared to those failing to implant. Future efforts should focus on associating specific metabolomic or proteomic signatures with normal patterns of epigenetic marks in in vitro culture models, which will greatly impact our ability to identify and generate embryos with high reproductive potential (reviewed in [155]).

As a concluding remark, NT cloning will have enormous benefits for basic research, preservation, and multiplication of desired genotypes, and for the generation of tailored animal models for human diseases.

## Figures and Tables

**Figure 1 ijms-22-00236-f001:**
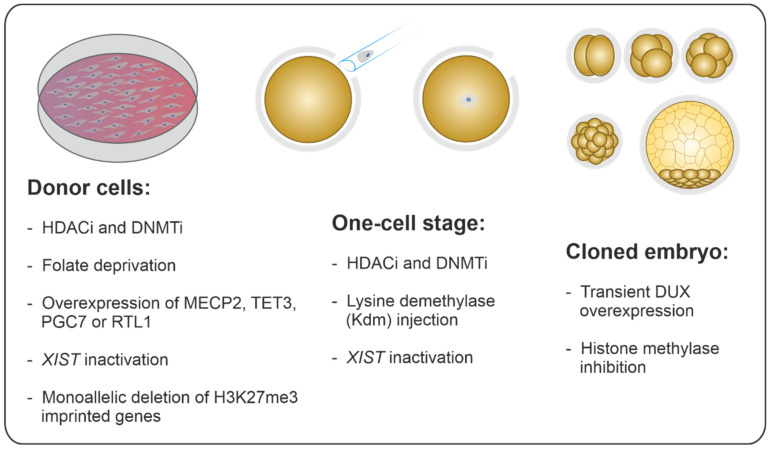
Epigenome Manipulation in Somatic Cell Nuclear Transfer (SCNT): possibilities to improve the outcome of SCNT experiments in the different steps of the procedure. HDACi: histone deacetylase inhibitors; DNMTi: DNA methyl-transferase inhibitors; MECP2: methyl-CpG–binding protein 2; TET3: tet methylcytosine dioxygenase 3; PGC7: primordial germ cell 7; RTL1: retrotransposon Gag like 1; XIST: X inactive specific transcript; H3K27me3: tri-methylation of histone-H3 lysine 27; DUX: double homeobox.

**Table 1 ijms-22-00236-t001:** Epigenetic barriers to somatic cell nuclear transfer cloning.

Epigenetic Barrier	Where	Reference
Memory of an active transcriptional state	Donor cells	[55,56]
Imprinting disorder in donor cells	Fibroblasts from abnormal cloned fetuses	[57]
Misregulation of mRNAs at the time of ZGA	Early stage NT embryos	[35]
Disturbed transcription by RNA polymerase I around ZGA	Early stage NT embryos	[38]
Non-proper degradation of maternally stored transcripts	Early stage NT embryos	[35]
Continuous expression of some somatic genes around ZGA	Early stage NT embryos	[40]
Resistance to reprogramming of pluripotency genes	Early to blastocyst stage NT embryos	[41,42]
Defective epigenetic reprogramming of DNA and histones	NT embryos	[27,46,58]
Abnormal regulation of DNA methyltransferase expression	NT embryos	[59]
Incomplete erasure of the somatic type of DNA methylation and somatic cell-like features	NT embryos	[40,51,60]
Failure to reactivate X chromosome and aberrant X chromosome inactivation (XCI)	NT embryos	[61,62]
Aberrant remethylation leading to mis-expression of genes and retrotransposons important for ZGA	NT embryos	[63]
Disruption of imprinted gene methylation and expression	NT embryos	[64,65]
Loss of imprinting	NT embryos	[53]
Defective trophoblast cell lineage specification	NT blastocysts	[66,67]
Abnormal gene expression profiles in cloned placenta	Extra-embryonic tissues	[68,69,70]
Abnormal imprinted gene expression and methylation patterns in mid-gestation	Cloned fetuses and placentas	[57,71,72,73,74,75,76]

## Data Availability

Not applicable.

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
