# Peer review of "Manipulating the Epigenome in Nuclear Transfer Cloning: Where, When and How"

_ijms, 2020, doi:10.3390/ijms22010236_

Round 1
Reviewer 1 Report
This manuscript is a very well written and comprehensive review on somatic cell nuclear transfer (SCNT) in animal cloning, documenting the problems encountered at the cell biology, biochemical, and molecular and epigenetic levels, the recent advances made to circumvent/modulate such problems, and future perspective such as the applications of CRISPR-dCas9 epigenome editing and other approaches to improve SCNT efficiency in animal cloning. A wealth of significant and relevant information can be found on epigenetic barriers to SCNT cloning, the role of DNA methylation, histone biochemistry and chromatin remodeling, X chromosome and the Xist gene, MECP2 protein, and genomic imprinting. The authors should also be given credited to cover the application CRISPR technology to modulate the epigenome in a temporal and spatial manner. This reviewer wishes to recommend that this manuscript merits acceptance for publication with high priority. The following minor editorial changes are suggested for the authors’ consideration for cosmetic reason:
P.6, line 214-217, the sentence should be - Gene expression and epigenome signatures of SCNT blastocysts from folate-deprived donor cells were more similar to blastocysts derived by in vitro fertilization than those of control SCNT blastocysts from folate-exposed donor cells [112].
P.8, Sfmbt2 on line 327 and SFMBT2 on line 329 – Are they the same gene and if yes, they should be referred to using identical nomenclature, i.e. either the same small or capital alphabets
Author Response
“P.6, line 214-217, the sentence should be - Gene expression and epigenome signatures of SCNT blastocysts from folate-deprived donor cells were more similar to blastocysts derived by in vitro fertilization than those of control SCNT blastocysts from folate-exposed donor cells [112].”
- We corrected the sentence as suggested.
“P.8, Sfmbt2 on line 327 and SFMBT2 on line 329 – Are they the same gene and if yes, they should be referred to using identical nomenclature, i.e. either the same small or capital alphabets”
- These are the same genes, but in line 327 “Sfmbt2” refers to the mouse, where following standard nomenclature gene symbols are italicized with the first letter in upper case and all the rest in lower case. In line 329, “SFMBT2” refers to pig, bovine and human, where gene symbols are completely in upper case and italicized. We followed the standard from the HUGO Gene Nomenclature Committee and the Mouse Genome Informatics (MGI) guidelines.
Reviewer 2 Report
This is a very informative and well-written article. I recommend publication after minor edits.
L40: into “an enucleated oocyte”
L73: consider changing “placement” by “integration”
L95: DNA packaging
L109: gene-repressive, but
L120: by the recipient
L188: reported
L298: “increased the blastocyst rate significantly increased” ? “significantly increased the blastocyst rate”
L349: or, in contrast, transcription
Author Response
L40: into “an enucleated oocyte”
- We added “enucleated” as suggested
L73: consider changing “placement” by “integration”
- We changed “placement” to “integration” as suggested. In our version of the manuscript, this is in line 74.
L95: DNA packaging
- We changed the sentence into “Histone proteins are key players in nucleosome formation and DNA packaging.” as suggested.
L109: gene-repressive, but
- In line 110, we added a comma after “gene-repressive”.
L120: by the recipient
- We added “the” in line 121.
L188: reported
- We changed “report” to “reported” in line 189.
L298: “increased the blastocyst rate significantly increased” ? “significantly increased the blastocyst rate”
- We corrected the sentence as suggested by removing “increased” once in line 299.
L349: or, in contrast, transcription
- We added commas in line 350.